# Designing and Distinguishing Meaningful Artisan Food Experiences

**Erin Percival Carter *** and **Stephanie Welcomer**

Maine Business School, University of Maine, Orono, ME 04469, USA; welcomer@maine.edu
* Correspondence: erin.p.carter@maine.edu; Tel.: +1-(207)-581-4944

**Abstract:** We examine consumer expectations about how specialty versus conventional food products affect well-being and how small, artisan producers can use that information to design better customer experiences. Drawing on recent work examining the costs and benefits of pleasure- and meaning-based consumption, we investigate whether consumer expectations that specialty products are more meaningful lead to increased desire for additional product information. We selectively sampled from the target market of interest: high-involvement consumers who regularly consume a food (cheese) in both more typical and specialty forms. The authors manipulate product type (typical versus special) within participant and measure differences in expected pleasure and meaning as well as a variety of behaviors related to and preference for additional product information. We find that these high-involvement consumers expect special food products to provide both more meaningful (hypothesized) and more pleasurable consumption experiences (not hypothesized) than typical food products. Consistent with our theory, consumer use of, search for, and preference for additional product information was greater for special products. A causal mediation analysis revealed that expectations of meaning mediate the relationship between product type and utility of product information, an effect which persists controlling for the unexpected difference in expected pleasure.

**Keywords:** food science; customer experience design; food well-being; food psychology; sustainability; hedonia; eudaimonia; meaningful consumption; artisan products; local food

## 1. Introduction

There is a growing movement among consumers to know more intimately where their food comes from and how it comes to be on their plate [1–3]. Providing consumers with information about how food was produced, efforts made to reduce negative environmental impacts, effects of local food purchases on developing rural economies, can each foster a deeper connection with food [4]. Consumers who value this connection often search outside of traditional outlets and product varieties to satisfy their needs [5]. For some, this can mean shifting their purchasing from grocery stores and chain restaurants to farmers' markets, harvest festivals, and restaurants specializing in local produce [6]. For others, this might mean trying hyper-local fare that does not fit in any traditional classification, or rediscovering old varieties and food traditions [7]. Consumers are motivated to engage in these costly behaviors at least in part due to the belief that consuming these different kinds of food will lead to improved consumer well-being [8].

Consumer researchers are also increasingly interested in better understanding the relationship between consumption and consumer well-being [9]. The term "well-being" can and has been used to refer to many different constructs: positive affect, life satisfaction, a sense of meaning, optimism, physical health, mental health, financial stability, the ability to regulate one's digital media environment and more. In recent years, consumer researchers have found it useful to distinguish between consumption motivated by a desire to maximize momentary, affective, hedonic pleasure-based aspects of well-being, and consumption motivated by a desire to cultivate a sense of purpose, connectivity, and the eudaimonic

meaning-based aspects of well-being [10–13]. Many of the factors that drive consumers to consume less conventional food or food from less conventional outlets appear to be more associated with pursuing purpose, connection, and meaning than affective pleasure yet the former has received less attention in the literature [14].

Building on this insight and recent work attempting to better understand meaningful consumption [11–13,15], in this paper, we provide information useful for scholars, as well as small-scale producers attempting to design specialty food products that cater to consumers' diverse needs. These producers and marketers currently face a dilemma. Should they generalize from best practices and research findings developed for more conventional producers competing based on economies of scale and with commodity products or should they seek to design products and consumption experiences better suited to the unique needs they satisfy? We suggest that smaller-scale agricultural producers are uniquely situated to meet consumers' unmet needs, particularly customers' desire to feel that their food provides not only pleasure but meaning. Building on recent work examining meaningful consumption, we analyze data collected from consumers who actively sought out local and artisanal products to examine the role of information in supporting meaningful consumption.

## 2. Literature Review

### 2.1. Consumption and Well-Being

What it means to live a good life has been a topic of discussion among philosophers, scientists, religious figures, and everyday people across the world and throughout history. In recent years, marketers and consumer researchers have paid increasing attention to better understanding how consumers think about their own well-being and how best to pursue a fuller, richer life [16]. For several reasons, not least of which were concerns about measurement validity [17], early work on this idea in consumer literature responded to the call for a field of hedonic psychology [18]. According to this tradition, well-being was best defined and measured as an integrated measure of one's affective experience [19–21]. The logic was that a life featuring relatively more positive affective experiences than negative was preferable and individuals' reported affect were more reliable over time than their reported life satisfaction; thus, identifying and communicating strategies for improving the affective quality of one's life would contribute to our understanding and promotion of well-being in a more reliable manner. Correspondingly, a seminal work in consumer behavior examining differences in motivations for consuming different types of products was Hirschman and Holbrook's [22] work distinguishing between hedonic experiences (characterized by the pursuit of pleasure and affective gratification) and utilitarian experiences (characterized by instrumental need fulfillment). The idea that the motivation for consumption can be categorized as either primarily hedonic or utilitarian has remained a bedrock assumption in much of the marketing literature [23–27].

There is, however, a class of experiences for which the hedonic versus utilitarian distinction seems to be inadequate. "Meaning" can be variously defined—we approach the concept of meaning drawing from a long history of scientific and philosophical work seeking to understand eudaimonia, the Greek word for living a full, meaningful, and deeply satisfying life and actualizing one's human potential [28–30]. Meaning is traditionally measured as subjective evaluations of how meaningful one's life is or the extent to which an experience contributes to one's sense of meaning [10,31,32]. Thus, people's lives and experiences can be said to be "meaningful" or "have meaning" to the extent that they contribute to one's sense of meaning [33].

Consumption, too, can be meaningful. Consumers motivated to pursue meaning select books, vacations, movies, concerts, restaurants, and other consumption experiences with an eye to which options they believe are likely to facilitate meaning making [11,12]. Importantly, pleasure- and meaning-based benefits should not be thought of as existing on one continuum but as orthogonal constructs. In other words, the same experience (eating a really delicious piece of local cheese) can provide a mix of utilitarian (eating this

makes me less hungry and I can get back to work without being distracted by hunger), pleasure (eating this is enjoyable and makes me feel happier right now) and meaning (I feel connected to the person who made this cheese and like I am learning something new) based benefits but the requirements and outcomes of each differ [12,13].

To illustrate, imagine three people who consume a piece of dark chocolate at the end of every day. Person 1 consumes chocolate every night because she saw a headline stating that doing so would reduce her risk of heart disease. She does not particularly enjoy dark chocolate, she prefers milk or white chocolate, but she understands that higher percentage cacao provides more significant health benefits. Each night she eats a piece of 85% cacao dark chocolate and then immediately brushes her teeth. This consumer is consuming primarily for utilitarian reasons.

Person 2 consumes chocolate every night as a kind of simple pleasure, a reward to himself for his efforts throughout the day. As he eats, he savors the flavor of the chocolate and the luxury of the moment. He enjoys this nightly ritual for those brief moments and then does not tend to think of it again until the next evening when he has another piece. He is consuming primarily for pleasure-based reasons.

Person 3 consumes chocolate every night because she likes to learn about the flavors, growing practices, and chocolate-making traditions and innovations across the world. She thinks very critically about fair trade standards, the chocolatiers who stay in business thanks to her purchases, and the uniqueness of the flavors in the chocolate that she samples each night. Eating chocolate makes her feel like she is connected to other people and she loves to reflect on how much she has learned about chocolate and the chocolate community over the years. Sometimes she eats a piece and realizes that the flavors are not necessarily her favorite but it does not really dampen the overall experience. Person 3 is consuming primarily for meaning-based reasons.

These vignettes together provide an extreme example of the ways in which the same product might contribute to a person's overall well-being in very different ways. It is important to note that often a consumption experience, particularly in the domain of food, will provide a mix of utilitarian, pleasure-, and meaning-based benefits. Indeed, work on the experiential pleasure of food (EPF) defines EPF as "the enduring cognitive (satisfaction) and emotional (i.e., delight) value consumers gain from savoring in multisensory, communal, and cultural meaning in food experiences [34]." This definition, therefore is consistent with our conceptualization but focuses instead on foods that provide both meaning-based ("enduring cognitive") and pleasure-based (emotional delight) benefits, whereas, in this work, we examine the unique implications of each type of benefit. Nevertheless, consumers can have primary motivations even when multiple benefits are possible. We hope that these vignettes can also help to build the intuition for the unique role that information plays in supporting meaning making.

Meaning making is reliant on expertise and supplementary information in a way that pleasure is not. While most people can enjoy hedonic benefits from consuming a piece of cheese or dark chocolate, the ability to derive meaning from these and other experiences is enhanced by baseline expertise about the product category or supplementary information provided at the time of consumption [12]. This is because while pleasure is a momentary affective assessment (i.e., asking oneself "Am I enjoying myself?"), meaning making is a cognitive process (i.e., asking oneself "What does this mean or change in a meaningful way about me or the way that I think about myself in relation to the world?"). While product knowledge and expertise have little or no effect on a consumer's ability to determine whether she feels relatively good or bad in a moment, they do affect her ability to integrate the consumption experience into her self-concept and her subjective evaluation of whether the experience contributes to a purposeful and self-actualized life. Drawing on this insight, in this work, we examine consumers' lay beliefs [35] about the role of product information in facilitating pleasure and meaning from food, particularly specialty or artisan food produced by small-scale producers. Our focus on lay beliefs

about well-being and their implications for consumption is consistent with recent work on meaningful consumption [11–13].

### 2.2. Unique Challenges of Small-Scale Farmer Producers

The agricultural sector in highly-developed nations has seen a dramatic shift in the composition of both the competitive landscape and the level of specialization necessary to compete in conventional agricultural markets over the past several decades. In both the US and Europe, the number of farms has dramatically decreased and the size of individual farms has increased as farmland and food processing has become increasingly industrialized and commodified [36,37]. Smaller-scale and diversified operations are being pushed out as the profitability of such operations decreases; in 2018, the median farm income in the US was a loss of $1840 [38].

Small-scale farms trying to make it in this climate find themselves faced with a choice: continue to sell their products into the commodity market where the primary competitive advantage is economies of scale or begin to market their products as non-commodity or value-added products [39–41]. While this strategy has the potential to increase profitability for small-scale producers who choose to remain small, it also demands different knowledge, resources, and abilities [42]. What commodity markets lack in flexibility and profitability, they make up for in terms of offering a simplified selling process. When a small-scale farmer shifts from streamlined, B2B, largely pre-determined sales processes to the much more abstract and nuanced B2C market for non-commodity and value-added products, she is forced to perform all of the functions of a major corporation with infinitesimal fractions of the time, resources, or experience. Recent research has demonstrated that this gap in knowledge, skills, and resources necessary to uncover and creatively meet consumer needs is a primary concern of small-scale farmers and craft and artisan food producers [43–45].

Small-scale farmers and craft and artisan food producers are uniquely suited to creatively approach their product design and positioning decisions. Smaller, more nimble producers are better able to ideate, prototype, and implement in quick succession. Small-scale producers who know both the production process and the customer intimately should be able to leverage this unique knowledge effectively. Finally, many small-scale agricultural and craft and artisan food producers were drawn to the work that they do for the same or similar reasons that consumers search for unconventional food products—a desire to live and consume more natural products, a desire to honor the past and preserve agricultural traditions, an urge to live more holistically and healthily. These producers may be able to better empathize with their target markets. Consumers increasingly pursue products not because of the way the product specifically performs but because of the experience that the consumer has during consumption [46,47].

### 3. Hypothesis Development

In formulating our hypotheses for this study, we drew on work demonstrating that consumers value and pursue meaningful consumption experiences and that these experiences are improved when consumers receive additional information about the product before the consumption experience [13]. Our goal in this study was to build on the observational insights of small-scale agricultural producers and craft and artisanal food makers to provide evidence and guidance on how consumers distinguish between their more specialized offerings and more typical food products, particularly with regard to effects on well-being. To do this, we examine consumer expectations about how specialty versus conventional food products affect well-being and how small, artisan producers can use that information to design better customer experiences. Thus, we offer the following hypotheses:

**Hypothesis 1 (H1).** *Consumers will report buying and consuming specialty products is more associated with the pursuit of meaning-based benefits than buying and consuming conventional products.*

We also wanted to provide insight that smaller-scale producers could use to design more empathetic consumption experiences that support the pursuit of meaning by examining consumer preferences for information about special versus typical food products. Thus, we offer the following three hypotheses:

**Hypotheses 2 (H2).** *Consumers will report that they use information on products labels more often when purchasing specialty products than conventional products.*

**Hypotheses 3 (H3).** *Consumers will report that they seek out additional information about a product from sources other than the product label more often when purchasing specialty products than conventional products.*

**Hypotheses 4 (H4).** *Consumers will report that their purchase and consumption experience would be improved with access to more information than is typically provided on the label to a greater degree for specialty than for conventional products.*

Finally, we tested whether our data support our hypothesis that expectations of meaning serve as the mechanism explaining differences in the value of product information for specialty and conventional products. We thus offer Figure 1 as a representation of our proposed process and the following mediational process hypothesis.

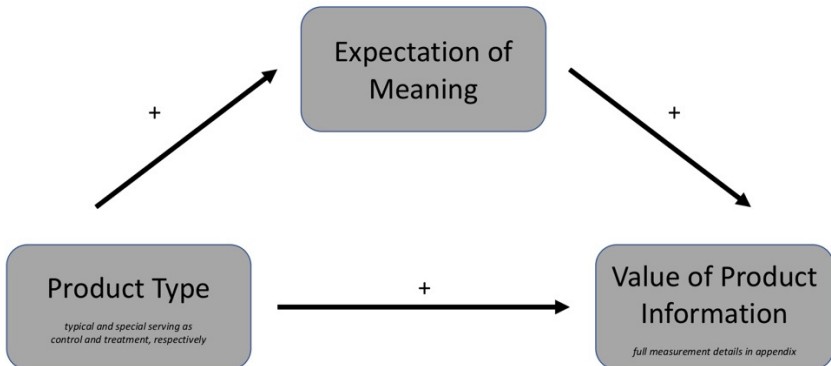

**Figure 1.** Conceptual model—consumer expectations of meaningful consumption experiences mediate the effect of product type on the value of product information.

**Hypothesis 5 (H5).** *The effect of specialty versus conventional product type on use of product label information will be mediated by expectations of meaning.*

**Hypothesis 6 (H6).** *The effect of specialty versus conventional product type on tendency to look for additional product information other than that included on the label will be mediated by expectations of meaning.*

**Hypothesis 7 (H7).** *The effect of specialty versus conventional product type on expectations of improved purchasing and consumption experiences resulting from more product information will be mediated by expectations of meaning.*

## 4. Materials and Methods

We surveyed high-involvement consumers of a food that is commonly consumed in more typical and more specialized forms and varieties: cheese. "Involvement" is a concept in the marketing literature which refers to an individual difference in the cognitive and emotional resources a consumer dedicates to thinking about and interacting with a given product class [48]. The literature often distinguished between enduring (more stable) and situational (affected by one's temporary environment) involvement [49]. We believe that consumers in our study were high in both enduring and situational involvement at the time of this study. Measuring within subject, we contrast consumers' expectations about the implications of consumption on well-being for typical and specialized versions of the same

product and examine the role of information in designing more compelling consumption experiences for each product type.

The study materials and procedure were reviewed and approved by the authors' Institutional Review Board. This study was conducted at a food festival in the Northeastern United States focused on artisanal cheese produced in the state in which the festival was held. The event was sponsored by the formal guild of cheesemakers in the state and members of the guild comprised the majority of the vendors in attendance at the event. Tickets to attend the festival ranged from $20 to 35 and the festival included cheese-tasting events and informational classes, live music, and access to a variety of artisan foods including cow, goat, and sheep's milk cheeses and yogurts as well as goat's milk caramels, spices, beer, and wine. We believe that the festival attendees are representative of the target market that small-scale dairy farmers and artisan cheesemakers target with their products (we are assuming psychographic segmentation based on the value consumers place on purchasing local, sustainable, and artisanal goods and not a segmentation based on demographic characteristics). Participants (N = 150, 66% female, average age = 43 years, median annual household income $50,000–59,999) participated in this study in exchange for a chance to win one of five prizes (choice of either $25 in cash or $25 worth of cheese selected by participating cheesemakers, though many participants indicated that they completed this study simply to help the members of the guild of cheesemakers). Study participants completed this study using paper and in an area the authors coordinated with the event sponsor to have set aside specifically for data collection.

In this study, we asked participants to think about "typical" and "special" cheese purchases (scales were developed by the authors; full descriptions and instructions are available in Appendix A). We used this terminology to limit the extent to which participants might think that we might have favored particular varieties of cheese, farms, or production methods; our goal was to contrast the products of small-scale producers with the more typical products of large-scale producers. We wanted to allow participants to define what made small-scale producers' products special in whatever way felt most natural to them.

We conducted all analyses using *R*. All effects were measured within subject. We examined whether the order of presentation of the special and conventional measures had any effect on our results and found none. We thus report the results of the simplified models for each hypothesis; all results are consistent and there are no changes to the significance of results when we include order of presentation in the models.

## 5. Results

### 5.1. Analyses

We present our findings in three parts. First, we examine consumer beliefs about the effect of typical and special products on well-being.

We first tested H1, that consumers would report buying and consuming specialty products is more associated with the pursuit of meaning-based benefits than buying and consuming conventional products. We did not have a prediction about the effect of product type on experienced pleasure. Participants indicated that products they deemed special were more likely to lead to both pleasurable ($M_{Special} = 4.59$, $M_{Typical} = 3.68$, $t(147) = -10.43$, $p < 0.0001$) and meaningful ($M_{Special} = 4.00$, $M_{Typical} = 2.87$, $t(145) = -9.96$, $p < 0.0001$) consumption experiences. While we did not expect the significant difference in expectations of pleasure, it is not inconsistent with out conceptualization which treats pleasure and meaning as orthogonal constructs. Our vignettes described extreme consumer types motivated by the pursuit of utilitarian, pleasure, or meaning-based needs but consumers can be motivated to fulfill multiple needs in a single purchase.

Our further hypotheses, however, were developed with an expectation that the difference in consumer expectations about the psychological benefits of specialty and conventional products would be particularly strong for meaning. Thus, as a follow-up analyses, we tested the magnitude of the product type difference for pleasure and meaning. We expected to find that the magnitude of the difference for our predicted effect on meaning

would be greater than the magnitude of the difference for the effect we did not predict on pleasure. In other words, we expected the difference in meaning for special versus typical products would be greater than the difference in pleasure for special versus typical products. We created new variables by subtracting the typical ratings from the special ratings for meaning and for pleasure for each participant and then compared the two difference measures. Consistent with our expectations, the difference between special and typical product experiences was significantly larger for meaning than for pleasure ($M_{meaning}$ = 1.13, $M_{pleasure}$ = 0.90, t (145) = 2.42, $p$ = 0.02.

Building on this insight and recent research examining the role of knowledge and expertise in meaning making, we next examined how consumer needs for and reactions to product information differ for more typical versus special food products. We examined responses to items measuring consumer use of, search for, and value of information regarding special versus typical food. Paired samples t-tests comparing special versus typical food revealed that special foods were associated with greater use of ($M_{Special}$ = 3.71, $M_{Typical}$ = 3.06, t (149) = −6.53, $p$ < 0.0001), search for ($M_{Special}$ = 3.01, $M_{Typical}$ = 2.05, t (150) = −10.66, $p$ < 0.0001), and value of ($M_{Special}$ = 3.38, $M_{Typical}$ = 2.82, t (148) = −7.21, $p$ < 0.0001) information than were typical foods. Note that degrees of freedom for the t-tests vary due to incomplete data on some items. All valid pairwise comparisons were utilized in each analysis. Removing participants with incomplete data from all analyses does not affect the pattern or significance of results. Across all three measures concerned with the utility of additional information about products, participants were more interested in information about special products.

Finally, we examine whether the expectation that a consumption experience will prove meaningful mediates the effect of product type on participants' use of, search for, and expected value of product information and find support for our theoretical model. We expected to find that the increased preference for information about special versus typical products would be driven at least in part by the expectation that special products were more likely to prove meaningful. However, we wanted to provide a conservative test of our theory given that the effect of product type on pleasure showed a similar pattern of results to the effect it had on meaning, the former of which we did not predict. Thus, we report our results below controlling for the effect of expectations of pleasure and examining only the unique contribution of expectations of meaning. The pattern and significance of all effects we report here is identical and the proportion of the total effect mediated is larger when we exclude pleasure from the models but we feel that this more conservative test is a better test of our proposed theoretical model.

We conducted our mediation analyses using the causal mediation analysis [50] which allowed us to account for the within-subjects random effects that result from our within-subject manipulation of product type and repeated measures of expected meaning, expected pleasure, and preference for information. The confidence intervals for each model were approximated utilizing a quasi-Bayesian method and 10,000 simulations of the data. The results of models predicting each measure of the value of product information supported our theoretical model and suggest a pattern of partial mediation; the indirect effect of product type (treating typical as control and special as treatment) through expectations of meaning was significant when predicting use of (indirect effect estimate = 0.163, 95% CI [0.08, 0.26], $p$ < 0.001), search for (indirect effect estimate = 0.164, 95% CI [0.08, 0.27], $p$ < 0.001), and value of (indirect effect estimate = 0.089, 95% CI [0.03, 0.17], $p$ = 0.004) information. Importantly for our theory, the pattern of effects does not hold if we control for meaning and examine pleasure as a mediator ($p$'s = 0.68, 0.18, and 0.66). Full results of all mediation analyses are available in Appendix B (Table A1).

## 5.2. Study Limitations

While we expect these results to generalize beyond these foci, it is important to note that our data used only a single product category (cheese) and collected data in one region (the Northeastern United States). It is also important to note that in our study, we sampled

on a factor we expect is necessary for these effects to emerge: a moderate to high degree of involvement. While we expect that these results would generalize to consumers that place at least a moderate value on the pursuit of meaning and have an interest and appreciation for artisan and specialty foods, we similarly expect that the results would not emerge among consumers with less motivation to pursue meaning or involvement with specialty and artisan foods. Critically, our data collection was limited to participants for whom it can reasonably be argued, the product category is of greater than average personal interest and importance. Participants in this study travelled and paid a minimum of $20 to enter a festival specifically focused on cheese. While this sample is not representative of the average consumer, we pursued this narrow sampling because product involvement is a theoretically important moderator for the effects that we investigated; a sample containing lower involvement consumers would likely find moderation of the effects found here. However, studies focused specifically on the judgments and decisions of this type of niche market are extremely limited and we believe that our unique sample allows for a novel and valuable contribution to our understanding of diverse consumer preferences.

One potential moderator of the effects we investigated in this paper unrelated to consumer involvement that should be considered in future work is the interaction between package labeling and retail outlet. Many of the people in this study were completing this study in a context in which they had direct access not only to the cheesemaker but also to the dairy producer—in many cases, this was the same person. Yet, past research has shown that consumer reaction to label content for sustainable product claims (specifically organic labeling) varies by retail outlet [51]. Consumer trust in information that appears on a label may be moderated by retail outlet, by the attributed source of the information [52], or by perceived distance in the value chain between the cheesemaker and the final consumer.

## 6. Discussion

In this paper, we examine how consumer expectations of the effect of typical versus special food products on distinct aspects of well-being differ and how the value of information differs for special vs. typical food products. We hypothesized and found that special food products are more associated with meaning than typical products and that when consumers encounter special food products they are likely to search for, value, and use information at higher levels than they would for typical products in part to cultivate meaningful consumption experiences. For producers of special products (such as artisan cheesemakers), this is notable, suggesting that they need to carefully design and promote the information accompanying the product and consumption experience.

### 6.1. Theoretical Contribution

Our investigation joins the growing body of work examining how consumers think about the relationship between different types of consumption and distinct aspects of their own well-being. We believe this work also contributes to a growing body of work in food psychology, agroecology, and sustainability that has identified the unique barriers to entry and long-term profitability for smaller and artisan producers [53] and has appealed to the growing interest among consumers in sustainable products as an inevitable part of the solution [54]. As this body of work continues to develop, it is critical that researchers consider more than measures of intention to purchase as there is often a gap between intention and behavior [55]. Decades of research on consumer behavior has shown us that a deeper and richer understanding of the mechanisms driving consumer judgements, decisions, and behaviors allows us to better predict and influence purchase behavior. We believe this work is an interesting case study of theories developed and mechanisms identified in the consumer literature [11–13] and an extension of that work to the broader discussion in food psychology and sustainability.

### 6.2. Managerial Implications

We believe that the managerial implications for this work for small-scale and artisan producers are even more significant. So many of these producers find themselves overextended, trying to manage legal, operational, entrepreneurial, and growth concerns on a day to day basis, often with a single employee and lacking formal education on all relevant aspects of their businesses and support to address structural barriers to their success [56]. Even when producers find the time to consult the relevant and actionable literature on marketplace trends and best business practices, that literature is very rarely developed with businesses at their scale in mind; instead, much of the focus is on markets for commodity goods or on larger-scale businesses. Much of the work that is focused on smaller-scale and artisan producers is focused on identifying barriers and reasons to explain the slow extinction of small family farms and other small agricultural enterprises. Instead, we focus on the unique opportunity that small-scale and artisan producers have to use their unique and intimate access and control over the story of production to design distinctive and meaningful consumption experiences.

Determining how best to tell the story of artisan production requires careful thought and consideration. In her extensive examination of artisan cheese, Heather Paxson refers to the complex cross-section of economic values and social values that underlie small-batch cheesemaking, "economies of sentiment" and points out that these economies, "point to the cultural, emotional, ethical, and political dispositions that motivate people to assume the economic risk and backbreaking labor of making cheese in small batches using minimal technology. These sentiments are multifaceted [57]." Intrinsic to economies of sentiment is an understanding of the values motivating the producer to perform the work, and these values largely connect production to wider social and ecological spheres. Paxson for instance found that artisan cheesemakers pursue this work for reasons including producing high-quality products, preserving local markets, supporting the dairy industry, and creating environmentally, socially and financially sustainable businesses [57]. With inspiration, the producer's main aim is to creatively explore both these values, resultant ties to the product's attributes, and how the consumer's experience can be more fully realized through information.

Implications of this research for specialty and artisan producers are clear; make additional information that can bolster meaning making available to consumers interested in meaningful aspects of consumption. Our items in this study were written with impersonal information in mind (i.e., product labels, websites, etc.). This is in part for efficiency; once developed and tested, these materials are easily reproduced without further taxing the time of the small-scale producer. Similarly, other research has suggested that while fostering connection between producers and consumers of local food contributes to the perceived value of local food, those connections may be more effective when the connection is indirect [58]. Producers should consider introducing supplemental information through a variety of channels. For instance, cheesemakers could use cards near their cheese displays that relay the provenance of the cheese—its sourced animal, date of milking, process used, and aging—this channel could be augmented with personal stories told by the cheese monger (when the cheesemaker is not directly involved in sales). Cards and information could be printed as takeaways, or consumers could be directed to a website, podcast, video series, or social media account with more detailed information. The ability to report granular, personal information of this nature is unique to smaller-scale producers and these producers should take advantage of the unique opportunity to cultivate meaningful consumer experiences this provides. Both producers and future academic research should further investigate a variety messages and media to determine the most effective messaging and outlets. As consumers' preferences evolve, small-scale producers need to adopt approaches to managing their products that help them to identify these unique opportunities for differentiation and increased profitability.

**Author Contributions:** Both authors contributed substantially to this project. E.P.C. and S.W. conceptualized this study, collected data, contributed to the writing, and secured funding. E.P.C. conducted all analyses using *R* and wrote the first draft of the paper. All authors have read and agreed to the published version of the manuscript.

**Funding:** This research was supported by the Competitive Energy Services Sustainability Fund managed by the Maine School of Business at the University of Maine. The fund provides curriculum support to promote economic development while protecting ecosystem health and fostering community well-being.

**Institutional Review Board Statement:** The study was conducted according to the guidelines of the Declaration of Helsinki, and judged exempt by the Institutional Review Board of the University of Maine (approved on 5 September 2019).

**Informed Consent Statement:** Informed consent was obtained from all subjects involved in the study.

**Data Availability Statement:** Anonymized data are available upon request, consistent with the Institutional Review Board review and approval of this project.

**Acknowledgments:** The authors thank the Maine Farmland Trust, the Maine Cheese Guild, and the 2020 class of Business, Agriculture, and Rural Development (BARD) Sustainable Business Fellows participating in the Technical Assistance pipeline program through the University of Maine for inspiring and supporting this work.

**Conflicts of Interest:** The authors declare no conflict of interest.

## Appendix A

*Appendix A.1. Descriptions of "Typical" and "Special" Cheeses That Appeared Earlier in the Survey*

For the next set of question, we'd like you to stop and think about a very typical cheese buying experience and what it means to you. Think about cheese that you purchase and eat frequently as well as where you typically purchase the cheese.

For the next set of question, we'd like you to stop and think about how you go about buying "special" cheese and what it means to you. Special means different things to different people; it might mean for a special occasion, it might mean something you've never had, it might mean something that you need for a specific recipe. There are no right or wrong definitions of special, we are interested in your opinions and experiences about what makes a particular cheese special to you.

*Appendix A.2. Items Measuring Use of, Search for, and Value of Information*

How often would you say that you closely read the labels to try to get more information when making typical and special cheese purchases?

→ Typical: five point scale from never to always
→ Special: five point scale from never to always

For each type of purchase, how often do you look for more information about the cheese from somewhere other than the label/packaging on the cheese itself (cheesemaker, cheesemonger, magazines, look online, ask friends, etc.)?

→ Typical: five point scale from never to always
→ Special: five point scale from never to always

To what extent would you say that your experience buying and consuming each type of cheese would be improved if you had more information about the cheese than what it typically provided on the package?

→ Typical: five point scale from never to always
→ Special: five point scale from never to always

*Appendix A.3. Items Measuring Effect on Well-Being*

To what extent would you say that your experience buying and consuming each type of cheese is pleasurable?

→ Typical: five point scale from "buying and consuming is not at all pleasurable" to "buying and consuming is extremely pleasurable"

To what extent would you say that your experience buying and consuming each type of cheese is meaningful?

→ Special: five point scale from "buying and consuming is not at all meaningful" to "buying and consuming is extremely meaningful"

**Appendix B**

**Table A1.** Full causal mediation results.

| Testing for mediation of the effect of product type on use of information | | | |
|---|---|---|---|
| | Simple model with meaning as mediator | Conservative model with meaning as mediator controlling for pleasure | Pleasure as mediator controlling for meaning |
| Indirect Effect | 0.394, CI = [0.27, 0.54], $p < 0.001$ | 0.163, CI = [0.08, 0.26], $p < 0.001$ | −0.013, CI = [−0.08, 0.05], $p = 0.68$ |
| Direct Effect | 0.59, CI = [0.39, 0.79], $p < 0.001$ | 0.605, CI = [0.39, 0.81], $p < 0.001$ | 0.605, CI = [0.40, 0.81], $p < 0.001$ |
| Total Effect | 0.99, CI = [0.80, 1.17], $p < 0.001$ | 0.767, CI = [0.55, 0.98], $p < 0.001$ | 0.59, CI = [−0.14, 0.09], $p < 0.001$ |
| Testing for mediation of the effect of product type on search for information | | | |
| | Simple model with meaning as mediator | Conservative model with meaning as mediator controlling for pleasure | Pleasure as mediator controlling for meaning |
| Indirect Effect | 0.396, CI = [0.27, 0.54], $p < 0.001$ | 0.163, CI = [0.08, 0.27], $p < 0.001$ | 0.039, CI = [−0.02, 0.10], $p = 0.18$ |
| Direct Effect | 0.593, CI = [0.39, 0.80], $p < 0.001$ | 0.607, CI = [0.40, 0.82], $p < 0.001$ | 0.321, CI = [0.13, 0.51], $p = 0.001$ |
| Total Effect | 0.99, CI = [0.80, 1.18], $p < 0.001$ | 0.771, CI = [0.55, 0.99], $p < 0.001$ | 0.360, CI = [0.17, 0.55], $p < 0.001$ |
| Testing for mediation of the effect of product type on value of information | | | |
| | Simple model with meaning as mediator | Conservative model with meaning as mediator controlling for pleasure | Pleasure as mediator controlling for meaning |
| Indirect Effect | 0.395, CI = [0.27, 0.54], $p < 0.001$ | 0.089, CI = [0.02, 0.17], $p = 0.004$ | −0.013, CI = [−0.08, 0.05], $p = 0.66$ |
| Direct Effect | 0.594, CI = [0.39, 0.80], $p < 0.001$ | 0.364, CI = [0.12, 0.60], $p = 0.003$ | 0.606, CI = [0.39, 0.81], $p < 0.001$ |
| Total Effect | 0.989, CI = [0.80, 1.17], $p < 0.001$ | 0.453, CI = [0.22, 0.69], $p < 0.001$ | 0.593, CI = [0.388, 0.79], $p < 0.001$ |

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
