# Peer review of "Designing and Distinguishing Meaningful Artisan Food Experiences"

_sustainability, doi:10.3390/su13158569_

Round 1

Reviewer 1 Report

Dear Authors, Dear Editor,

The authors analyse those purchasing behaviours that are very important in terms of sustainable food consumption: choosing quality products, including certified ones, and local products, including artisanal and small-scale production. By analysing meaningful consumption, the authors examined to what extent it is driven by eudaimonic and to what extent by hedonistic motives.

The manuscript is a valuable study, but I have some major comments/suggestions:

1. Title. I suggest including the word cheese, e.g. ...experiences with/for/around etc. cheese; this addition may increase the audience.

2. Keywords. Food science, positive psychology, sustainability are inadequate to the content; well-being is unnecessarily double-indicated; words such as local/craft/special and typical cheese, pleasure,  eudaimonia, meaningful consumption are missing.

3. Purpose. It is stated three times and each time in a different way: in the abstract (L. 7-9), in the last paragraph of subsection 1.1. (L. 181-185) and in the last paragraph of subsection 1.2. (L. 229-243), where it is additionally mixed with hypotheses. It is necessary to state the essential aim clearly in the abstract and to develop it in detail before chapter 2. Then the research hypotheses, comprising the authors' theory, which are scattered in the Introduction and in the Results chapter, should be listed.

4. Materials & Methods. Determining the location of the study was a good idea to reach consumers interested in meaningful consumption. It is necessary to supplement the information regarding the research material by indicating the state (Illinois?) and city where the food festival took place (Chicago?), because the US both as a country (geography, economy, etc.) and a community are wildly diverse. In this context, more socio-demographic characteristics of the respondents are worth presenting.

The chapter does not provide any information on the methods of analysis.

5. Results. I propose to move the descriptions of the methods of analysis from here to Chapter 2, and just use the names of the methods here. In paragraph L. 326-338 the authors provided a critical reference to the limitations of their study, but also to its strengths. I propose to exclude this paragraph and place it after the Discussion. In turn, paragraph L.340-348 should be moved to the Discussion section. Certainly through forgetfulness the authors have left lines L.350-351 in the manuscript. Figure 1 should go to its proper place in the manuscript and needs a minimum of explanation.

6. Discussion. It is quite modest, the authors refer only to one book, concerning the situation from a decade ago, which is an epoch in the context of changes on the food market or more broadly in global food systems, and they quote an interview with the owner of a retail outlet. The discussion needs to be supplemented concerning attitudes and motives for choosing special/quality foods, explained by the nutritional psychology of modern consumers. Such research is conducted in the US, in European and other highly developed countries. It is also worth presenting the results of the study in the context of the principles of sustainable food consumption.

Minor comments:

L. 189 - the sentence should be completed that it refers to the agricultural sector in highly developed countries;

L.270 - the value for Mspecial should be given with two decimal places (4.00). 

Kind regards.

Reviewer 2 Report

The article contains very important and up-to-date content in the field of hedonic psychology. More and more consumers' attention is focused on the pleasure of consumption.
The research conducted for the purpose of this article has some limitations, but it is important that the Authors noted these limitations.
There are editorial errors in the article.

Reviewer 3 Report

The topic of this paper is interesting and the test product is well selected. Please consider my comments and suggestions, which are here listed.

  • The third paragraph in the Introduction chapter – from line 50 – has no citation.
  • Hypotheses of the paper are hidden in the text body. I recommend authors highlight their Hypotheses add numbers for them and – most importantly – give background to all of the Hypotheses.
  • The fourth paragraph in the Introduction chapter – from line 64 – should be rather a part of the Methodology chapter. I recommend authors place this part in the Methodology chapter.
  • The authors mention in line 69 the Figure 1, which can be found at the very end of the paper. I do not understand its reason.
  • Authors write in the Introduction chapter that “We find (…)” and “We conclude (…)”. These seem to be results and I don’t understand why are findings in the Introduction chapter. Also, in line 241 authors add “we find (…)”…
  • The authors describe typical consumers from line 130. I think authors should give some pieces of evidence that these are the consumer types. Now it seems to rely only on their own observation. Anyway, I don’t think these examples best fit the place where they are now in the paper.
  • From line 166 to line 168 authors add a research question to the Introduction chapter. I think research questions and hypotheses should be placed together in the paper e.g. at the end of the Introduction chapter.
  • In lines, 174-177 authors add two questions. It is not clear if they used these in their research. If they did, why are these questions in the Introduction chapter and not in Methodology? If these are validated scales readers should know the source of these scales.
  • Also in the line of 180, the authors state that “in this work we sought to examine consumers’ lay beliefs (…)”. This is a research goal that should have a clear background.
  • In line 233 Authors state that “we predicted that consumers (…)”. So, it seems to be a hypothesis without background. Please add relevant literature to this.
  • The methodology chapter is quite short. The methodological background of this paper could be deeper. Some methodological effort would significantly increase the strength of the paper.
  • The authors used questions and scales for their work. It is not clear if these scales were developed by authors or they used validated scales.
  • I think the paragraph from 326 is rather the Limitation part of the paper and this should not be placed in the Results chapter.
  • The authors left the default statement in line 351. “All figures and tables should be cited in the main text as Figure 1, Table 1, etc.” Anyway, I don’t think that this is the best place for Figure 1.
  • Two names appear in the Discussion chapter. I think these names and their view do not fit the Discussion chapter. Also, I am not sure that a blog is a good source for a scientific paper. If the Authors think these help their paper I recommend them to place these statements in the Introduction chapter.

Reviewer 4 Report

Overall a well presented article.  I pose some questions below.

Lines 26-27: Making this opening statement should be supported by a citation.

Line 37: Is this the consumers’ well-being?

Line 41: With many different constructs, should you list some?

Line 41: Which researchers have found it useful?

Line 64: What would entail being considered a high-involvement consumer?

Line 69: Since there is only 1 figure, it may be easier to read the article if the figure were close to the citation instead of in a separate section.

Line 91: Which researchers have paid increasing attention?

Line 101: Isn’t “the most influential” subjective?

Lines 130-153: A very good differentiation of consumer types

Lines 321-324: Can you comment on why you think the pattern did not hold up in this situation?

Lines 326-328: Can you actually translate these results to the average consumer?  Could it be possible that the consumers you observed were actually seeking meaning as evidenced by willingness to pay a gate fee to enter the festival?

Line 378: Would sustainability include profitability?

Lines 396-403: Why did this business close?  If it closed because it was not successful or profitable, can we trust the analysis presented by the former owner?

Reviewer 5 Report

Very interesting topic and very interesting research. Some recommendations to make the article stronger.

  1. Separate introduction from literature review. Include in the introduction the traditional components such as contextualization, research gap, research objectives, etc.
  2. Include more references in the contextualization (introduction) to make it stronger.
  3. In the literature review, the article has long paragraphs with the references usually in the end of it. Please cite your references along the paragraph, so it is clear what ideas are supported by whom. I would also avoid circunstancial writing such as the examples of chocolate consumption included in section 1.1 (or at least I would reduce it to 3-4 lines). 
  4. Research questions and/or hypotheses should be highlighted.
  5. Method section is poor. Please include more information not only on data collection, but also sampling, adequacy of the method to the research objectives/questions/hypotheses, and data analysis techniques.
  6. Discussion has almost no references - your results should "discuss" with the most relevant literature cited in the article.
  7. Please include a final section for conclusion, with the traditional components: theoretical contributions, implications for managers, limitations, future research directions. Some sections of text should therefore migrate from discussion to those new subsections.

Overall this is a very interesting article. 

One side note: I really like the title, but I'm not sure it is completely aligned with the article - it is too broad... good for keywords, but I'd prefer a more specific title.

Round 2

Reviewer 1 Report

The authors have achieved an excellent result after taking into account the reviewers' comments, including mine (while retaining the right to disagree with some). As it stands, the paper represents an appropriate scientific level for the journal. I have only two more minor comments: 1 - it is redundant to present the purpose of the paper in subsection 2.1. (L. 162-171), since chapter 3 has been extracted and the same information is contained therein, 2 - I propose to shorten the title of subsection 5.2. to "Study Limitations".

Kind regards.

Author Response

We thank you for the additional comments and again for the comments from the first round of review.

We have revised the section in 2.1 (though we note that it appeared in what appeared as lines 262-271 in the version of the paper that was provided through the download link for this round of revision). Our intent in adding that section within 2.1 had really been to introduce and explain the focus on consumer lay beliefs and we believe the changes that we have made maintain that introduction and explanation without the redundancy in stating the purpose of the paper.

We have also changed the title of section 5.2 following your suggestions.

Sincere thanks again for your comments throughout the review process.

Reviewer 3 Report

Thank you for your efforts to strengthen the paper and for following my recommendations.

Author Response

Thank you very much for the comments and suggestions. We feel the paper was much improved by this review process.